# Pathological Changes in Extracellular Matrix Composition Orchestrate the Fibrotic Feedback Loop Through Macrophage Activation in Dupuytren’s Contracture

**DOI:** 10.3390/ijms26073146

**Published:** 2025-03-28

**Authors:** Elizabeth Heinmäe, Kristina Mäemets-Allas, Katre Maasalu, Darja Vastšjonok, Mariliis Klaas

**Affiliations:** 1Institute of Molecular and Cell Biology, University of Tartu, Riia 23b, 51010 Tartu, Estonia; elizabeth.heinmae@ki.se (E.H.); kristina.maemets-allas@ut.ee (K.M.-A.); darja.vastsjonok@ut.ee (D.V.); 2Department of Cell and Molecular Biology, Karolinska Institute, 171 77 Stockholm, Sweden; 3Department of Traumatology and Orthopedics, Institute of Clinical Medicine, University of Tartu, 51010 Tartu, Estonia; katre.maasalu@kliinikum.ee; 4Clinic of Traumatology and Orthopedics, Tartu University Hospital, 51010 Tartu, Estonia

**Keywords:** fibrosis, Dupuytren’s contracture, macrophages, extracellular matrix, myofibroblasts

## Abstract

Dupuytren’s contracture belongs to a group of fibrotic diseases that have similar mechanisms but lack effective treatment and prevention options. The excessive accumulation of connective tissue in Dupuytren’s disease leads to palmar fibrosis that results in contracture deformities. The present study aimed to investigate how the tissue microenvironment in Dupuytren’s contracture affects the phenotypic differentiation of macrophages, which leads to an inflammatory response and the development of chronicity in fibrotic disease. We utilized a decellularization-based method combined with proteomic analysis to identify shifts in extracellular matrix composition and the surrounding tissue microenvironment. We found that the expression of several matricellular proteins, such as MFAP4, EFEMP1 (fibulin-3), and ANGPTL2, was elevated in Dupuytren’s tissue. We show that, in response to the changes in the extracellular matrix of Dupuytren’s contracture, macrophages regulate the fibrotic process by cytokine production, promote myofibroblast differentiation, and increase the fibroblast migration rate. Moreover, we found that the extracellular matrix of Dupuytren’s contracture directly supports the macrophage-to-myofibroblast transition, which could be another contributor to Dupuytren’s disease pathogenesis. Our results suggest that interactions between macrophages and the extracellular matrix should be considered as targets for novel fibrotic disease treatment and prevention strategies in the future.

## 1. Introduction

The extracellular matrix (ECM) is a complex intercellular structure in which macromolecules are assembled into a mechanical network in a tissue-specific manner. As a highly dynamic structure, the ECM also functions as a reservoir of growth factors and bioactive molecules, making it an important element that controls the most fundamental processes and properties of cells and tissues, such as proliferation, adhesion, migration, polarity, differentiation, and apoptosis [1]. The major components of the ECM include about 300 proteins, such as collagens, proteoglycans, elastin, and cell-binding glycoproteins, each with distinct physical and biochemical properties [2].

The ECM is subject to continuous remodelling, with its components undergoing reorganization, degradation, or alteration. In addition to intrinsic processes, it is also modulated by exogenous stimuli, such as cytokines, glucocorticoids, oxidative stress, pressure, and mechanical stretch [3]. Changes in the ECM composition are most pronounced during tissue regeneration, which is accompanied by collagen deposition and the induction of multiple matricellular proteins in response to tissue injury. ECM deposition is an indispensable and usually reversible part of the wound healing process [4,5]. However, if tissue damage is severe or chronic, or if the wound healing response is no longer effectively regulated, tissue repair can gradually become an irreversible fibrotic process [6].

Any fibrotic process is a cell–matrix interaction. In fibrosis, an excessive accumulation of ECM components, such as collagen and fibronectin, occurs in and around the inflamed or damaged tissue [7]. Fibrosis can lead to permanent organ scarring, dysfunction, and eventually, organ failure and death. Fibrosis has been described in a number of conditions and diseases, such as end-stage liver disease, renal disease, idiopathic pulmonary fibrosis, heart failure, many chronic autoimmune diseases including scleroderma, rheumatoid arthritis, Crohn’s disease, ulcerative colitis, myelofibrosis, and systemic lupus erythematosus [8]. All forms of fibrosis are characterized by chronic inflammatory–immunological reactions that occur in the early stages of the disease; these involve both the innate and adaptive immune systems and promote the development of subsequent fibrotic processes [9]. The stimulation of the cellular components of the innate immune system results in either the release of profibrotic factors or the production and secretion of newly synthesized factors, such as profibrotic cytokines [6,9]. Therefore, it is considered that fibrosis cannot occur without inflammation.

Dupuytren’s disease (DD) is a fibroproliferative disorder characterized by fibrosis of the palmar and digital fascia. The disease causes fingers to become permanently bent in a flexed position, with the 4th and 5th fingers of the hand most often affected. In Western populations, DD affects 12% of individuals aged 55, with occurrence rising to 29% among those aged 75 years and older [10]. Although the pathogenesis of DD has been studied for some time, the surgical resection of the diseased tissue remains the primary treatment option when the flexion deformities are already severely impairing hand function. In addition, there are no effective treatments to prevent DD progression or to reduce the recurrence rates [11].

A combination of environmental factors, especially mechanical stress and smoking, as well as genetic predisposition, appear to contribute to the pathogenesis of DD [11,12]. The exact mechanism by which inflammation occurs in DD still needs to be elucidated, but microvascular damage and occlusion are likely among the triggering events. As in other fibrotic diseases, the local production of chemokines plays an important role in the immune cell recruitment in DD. One of the main effector cell types is the myofibroblast, which is responsible for excessive matrix deposition, remodelling, and contraction [13]. Additionally, macrophages are considered to be important regulators of fibrosis [14]. Macrophages are permanently resident in all tissues; inflammatory reactions also stimulate monocyte migration and differentiation into macrophages in response to specific triggers. An accumulating body of evidence suggests that the microenvironment influences the phenotypic differentiation of macrophages by modulating cell activation and polarization, leading to the propagation of the inflammatory response and the development of chronic disease [15]. In addition to producing profibrotic cytokines, activated macrophages recruit myofibroblasts and exacerbate immune cell infiltration at the sites of tissue damage, thereby promoting tissue breakdown, excessive matrix deposition, and ultimately, fibrosis [16,17]. Current knowledge of ECM–macrophage interactions in fibrotic diseases remains limited and fragmentary; further research into this topic is required.

We investigated the matricellular ECM proteins upregulated in DD ECM. To characterize the role of ECM components in DD pathogenesis, we used a systemic proteomics-based approach. Previous studies conducted by our group [18] and other research groups [19,20] have shown that the decellularization of tissue is an effective method for enriching tissue samples with ECM-associated proteins for proteomic analysis. We identified several profibrotic components upregulated in DD ECM and analysed the signaling pathways potentially involved in DD. Moreover, we showed that fibrotic ECM-derived signals could contribute to macrophage-to-myofibroblast transition and directly promote the progression of fibrosis.

## 2. Results

### 2.1. Proteomic Profile of the Changes in the ECM of Dupuytren’s Contracture

Proteomic analysis was conducted to investigate the role of ECM in the pathogenesis of Dupuytren’s contracture. Tissue specimens from patients diagnosed with Dupuytren’s contracture and from control patients’ normal palmar fascia were decellularized to enrich ECM-associated proteins in the tissue samples (Figure 1A).

The proteomic analysis identified a total of 1167 proteins, of which 162 were differentially expressed between Dupuytren’s and control samples (FC > 1.2). The list of significantly regulated proteins is provided in Appendix A. Of the differentially expressed proteins, 98 were downregulated and 64 were upregulated in the DD group compared to the control group (Figure 1A). The analysis also considered the patient’s gender and age, yet no differences in protein expression were identified based on these factors.

The heatmap of results shows the clear clustering of DD and control patient samples into two distinct groups (Figure 1B). In Dupuytren’s contracture, the expression of upregulated proteins was uniformly elevated in all patients. However, greater variability was found among the downregulated proteins. The volcano plot (Figure 1C) shows seven upregulated proteins in Dupuytren’s contracture chosen for further investigation: chymase 1 (CMA1), von Willebrand factor A domain containing 1 (VWA1), microfibril-associated glycoprotein 4 (MFAP4), EGF containing fibulin extracellular matrix protein 1 (EFEMP1), fibrillin 1 (FBN1), periostin (POSTN), and angiopoietin-related protein 2 (ANGPTL2). The selection criteria for these proteins included a minimum 1.2-fold increase in their expression, as well as their established roles in other fibrotic diseases and in the inflammatory response.

### 2.2. Analysis of the Signalling Pathways Potentially Involved in DD

A KEGG pathway analysis was performed to investigate the pathways associated with differentially expressed genes (Figure 2). The analysis revealed that the upregulated proteins were involved in protein degradation (encoded by 16 genes) and ECM–receptor interactions (encoded by 12 genes). Most downregulated proteins were associated with complement cascade regulation (encoded by 11 genes), focal adhesions (encoded by 9 genes), and ECM–receptor interactions (encoded by 5 genes). Furthermore, a more thorough analysis of protein–protein interactions in DD divided the upregulated proteins into four major groups (Figure 3). Upregulated proteins in Dupuytren’s contracture were primarily responsible for angiotensin regulation, matrix metalloproteinase activation, elastic fibre formation, collagen trimerization, and ECM organization, suggesting that extensive tissue remodelling forms the molecular basis of the disease. Interestingly, the analysis revealed that the significantly upregulated MFAP4 can interact with both matricellular (FBLN2, FBLN5, MFAP2) and structural proteins (ELN, FBN1) (Figure 3). However, some of the upregulated profibrotic proteins, such as ANGPTL2, did not form associations with other upregulated proteins.

The group of downregulated proteins in DD tissues included several proteins with known antifibrotic properties, such as cartilage oligomeric matrix protein (COMP), proteoglycan 4 (PRG4), growth differentiation factor 10 (GDF10), bone morphogenetic protein 3 (BMP3), and annexin A1 (ANXA1) (Figure 1C). The downregulated proteins were divided into three clusters based on their primary functions, which included complement and coagulation cascade regulation and actin cytoskeleton regulation (Figure 2, Appendix A).

Next, we applied immunofluorescence analysis to Dupuytren’s contracture and control palmar fascia tissue sections to investigate the expression patterns of selected profibrotic proteins from the proteomic analysis (Figure 4). The quantification of the fluorescence of selected markers revealed that the expression of ANGPTL2 was, on average, 6.9 times higher (*p* < 0.0001) in DD tissue sections compared to control palmar fascia. MFAP4 expression was detected at a 4.3-fold higher level (*p* < 0.0001) and EFEMP1 at a 6.3-fold higher level (*p* < 0.0001) in DD tissue sections compared to control palmar fascia tissue sections (Figure 4). Furthermore, we detected FBN1 expression at 2.2-fold higher levels, POSTN at 2.9-fold higher levels (*p* < 0.001), and VWA1 expression at 7.3-fold higher levels (*p* < 0.001) in DD tissue sections compared to control palmar fascia (Appendix A). Consistent with previous studies [15], we noted an increased accumulation of macrophages in fibrotic tissues, as the expression of the macrophage marker CD68 was 3.6 times higher (*p* = 0.001) in DD tissue sections than in palmar fascia control sections (Appendix A). The accumulation of macrophages was particularly notable adjacent to ANGPTL2, MFAP4, and EFEMP1 depositions (Figure 4A).

### 2.3. The Effect of DD-Derived Extracellular Matrix on Macrophage Differentiation

Previous studies have found that macrophages are major players in fibrotic diseases, including Dupuytren’s contracture [12,21,22]. To investigate the effects of exposure to DD ECM on macrophage cytokine production, we incubated monocyte-derived macrophages with decellularized DD or control palmar fascia ECM. RT-qPCR analysis revealed the significant upregulation of *IL6* (*p* = 0.02), *TNF* (*p* = 0.03), *IL1B* (*p* = 0.01), and *IL10* (*p* = 0.03) expression (Figure 5A) after a 4 h incubation with DD ECM compared to the control ECM. However, the expression level of the well-characterized profibrotic cytokine TGF-β was not significantly increased. To confirm the production of cytokines at the protein level, we analysed the macrophage culture medium using ELISA (Figure 5B). We detected an 8.9-fold greater secretion of IL-6 (*p* = 0.006), a 4.6-fold greater secretion of TNF (*p* = 0.014), and a 3.0-fold greater secretion of IL-10 (*p* = 0.015). The levels of IL-1β and TGF-β remained below the detection limits.

Macrophages contribute to the development of several fibrotic diseases by chemokine and cytokine production, which promote fibroblast proliferation and differentiation into a critical ECM-secreting cell type, myofibroblast [21]. In recent years, a novel mechanism, macrophage-to-myofibroblast transition (MMT), has been identified as an important contributor to renal [23] and liver fibrosis [24]. To investigate whether MMT could contribute to DD pathogenesis, we analysed DD and control palmar fascia tissue samples (Figure 6). We observed the co-localisation of α-SMA^+^ and CD68^+^ cells, with some indication of overlap between the myofibroblast marker α-SMA and the macrophage marker CD68 in some of the cells located near the sweat gland-rich nodules. This suggests that MMT could play a role in the pathogenesis of DD.

To gain a further insight into the potential mechanisms of MMT in DD, we stimulated human monocyte-derived macrophages with decellularized homogenized DD or control palmar fascia ECM and analysed the expression of α-SMA using immunofluorescence imaging (Figure 7). We found that, in the presence of DD tissue-derived ECM, the macrophages significantly upregulated α-SMA expression (*p* = 0.0016), indicating that fibrotic ECM-derived signals could contribute to MMT and promote the progression of fibrosis.

### 2.4. The Effects of DD ECM on Fibroblast Differentiation and Migration

Next, we investigated whether DD ECM affects fibroblast differentiation toward the myofibroblast phenotype. The decellularized DD or control tissue ECM was directly added to the fibroblast culture, and the cell proliferation rate and changes in the expression of myofibroblast markers were observed using immunofluorescence microscopy. No changes in Ki67-antigen, α-SMA, or type I collagen expression were detected (Appendix A). As we could not show a direct effect of the fibrotic ECM on fibroblast-to-myofibroblast differentiation, we further investigated whether the role of ECM in fibrotic disease could be mediated through macrophages. To understand the role of macrophage–ECM interactions in the development of fibrotic disease, we first incubated macrophages with decellularized DD or control palmar tissue ECM and then added the macrophage culture medium to DD tissue-derived fibroblasts. After 48 h of culture in macrophage-conditioned medium, we found that cell proliferation increased 1.3-fold (*p* = 0.001) in DD ECM-stimulated macrophage medium compared to cells that were cultured in the control medium (Figure 8). In addition, we detected a 1.4-fold increase in type I collagen expression (*p* < 0.001) and a 1.6-fold increase in α-SMA expression (*p* = 0.04). These findings indicate that fibrotic ECM-derived signals influence the cells to differentiate toward an α-SMA-positive myofibroblast-like phenotype. These observations were supported by changes in fibroblast cell shape and increased stress fibre formation (Figure 8), which are characteristic of the myofibroblast transition process [25].

Macrophages play an active role in fibrosis, and their signalling can influence the behaviour of other cells. To study the role of macrophage-derived signals in fibroblasts, we investigated their function in regulating cell migration. After the macrophages were cultured in DD or control ECM-containing medium, we used this macrophage culture medium to stimulate the fibroblasts for 48 h and then quantified the cell migration rate (Figure 9). Compared with control medium-exposed fibroblasts, fibroblasts stimulated with DD-conditioned macrophage medium migrated 1.58-fold more effectively (*p* = 0.004) after 24 h of incubation in an 8 µm pore size transwell migration chamber. The effect of DD ECM-derived signals on fibroblast migration was similar to that of M1-derived macrophage signals, suggesting that polarized macrophages can promote fibroblast migration in fibrotic tissue.

## 3. Discussion

Dupuytren’s contracture (DD) belongs to a large group of fibrotic diseases that share similar mechanisms and currently lack effective treatment and prevention options. Unlike fibrotic diseases affecting internal organs, like the heart, lungs or liver, which are typically diagnosed at later stages when organ function is already irreversibly impaired, DD can be diagnosed early and is readily accessible for scientific research after surgical excision. This has enabled us to analyse the mechanisms of active fibrotic processes in the relatively early-stage nodular tissue ex vivo. While the cellular mechanisms of fibrosis have been extensively studied [6,8], the role of the tissue microenvironment and ECM in this process is still not fully understood. The previous proteomic profiling of DD tissue was carried out more than a decade ago [26]. Although the study by Kraljevic Pavelic et al. identified several major ECM components upregulated in DD, the majority of the matricellular proteins with lower expression levels remained undetectable.

Here, we have described the changes in the ECM-associated proteome of DD tissue and identified components that potentially activate or promote the fibrosis in DD. Furthermore, our results indicate that the pathological changes in the diseased ECM composition orchestrate a feedback loop through macrophage activation, which promotes myofibroblast activation and differentiation. In the present study, we, for the first time, have identified shifts in ECM composition and the surrounding tissue microenvironment using a decellularization-based method combined with a high throughput proteomics approach. The upregulated proteins in DD ECM were mostly associated with ECM reorganisation, and the analysis of the signalling pathways associated with the upregulated proteins in DD indicates that the major changes were linked to protein degradation and ECM–receptor interactions. These findings align closely with the well-established knowledge that ECM is extensively remodelled in fibrotic tissue [27].

Depending on the stage of the disease, the following two different matrix structures are distinguished in DD: a cell nodule and a matrix-rich cord. Nodules appear at an early stage of the disease and represent dense cellular aggregates with the foci of myofibroblasts. As the disease progresses, the nodules expand and form cord-like formations of relatively cell-poor, regularly organised, parallel collagen bundles, producing a linear structure extending to the fingers. It has been shown that, as the disease progresses, this collagen-rich cord is simultaneously degraded, and novel cross-linking bonds form, which lead to cord shortening and contraction [28]. The main functions of the upregulated proteins in our study were associated with matrix metalloprotease activation, the formation of elastic fibres, the trimerization of the collagen chains, and ECM reorganisation. These results indicate that the upregulated proteins and their interactions identified in this work could play an active role in DD pathogenesis.

Several matricellular proteins have surfaced in recent years as major regulators of closely related physiological processes, such as cutaneous wound healing [5] and fibrosis in the heart [29] and liver [30]. Here, we found that the expression of several matricellular proteins, such as MFAP4, EFEMP1, CMA1, FBN1, VWA1, POSTN, and ANGPTL2, was elevated in DD tissue. MFAP4 has been suggested to be actively involved in the pathogenesis of several matrix remodelling-associated diseases, such as liver cirrhosis, cardiovascular disorders, and cancer, through its interactions with integrins and proinflammatory signalling, as well as modulating the TGF-β pathway [31,32]. Proteomic analysis of liver tissue samples has identified upregulated MFAP4 in HBV/HCV-associated hepatic fibrosis [33], and several reports have suggested that serum MFAP4 levels can be used as a non-invasive tool for screening liver fibrosis [34,35]. Additionally, a cohort study has reported increased MFAP4 serum levels in pulmonary fibrosis caused by chronic obstructive pulmonary disease [36]. However, in a study involving patients with idiopathic pulmonary fibrosis and a mouse model of lung fibrosis, MFAP4 was not found to have a significant effect in the development of pulmonary fibrosis [37]. ANGPTL2 is a secreted glycoprotein that is thought to maintain tissue homeostasis and participate in tissue repair. An excess of ANGPTL2 causes chronic inflammation by activating NF-κB inflammatory signalling through integrin α5β1 [38] and irreversible tissue remodelling associated with the progression of metabolic diseases, such as obesity, type 2 diabetes, and atherosclerosis [39]. ANGPTL2 plays a significant role in atrial fibrosis [40] and in renal fibrotic processes via TGF-β signalling [41]. Conversely, in a mouse model of lung fibrosis, ANGPTL2 was shown to have a protective effect against the disease [42]. EFEMP1 (fibulin-3) is a secreted ECM glycoprotein, with a diverse array of pathophysiological associations [43]. Increased EFEMP1 expression has been reported in cardiac fibrosis, and it contributes to scar formation after myocardial infarction [44] by participating in elastic fibre assembly and promoting inflammation [44,45]. EFEMP1 has been found to regulate the organisation of elastic fibres in the extrahepatic biliary tract [46]. In addition, a genome-wide association study has linked EFEMP1 to biliary atresia, a fibrotic disease of the extrahepatic biliary tract in newborns [46,47]. As far as we are aware, there are no previous reports describing the role of MFAP4, ANGPTL2, or EFEMP1 in DD. Notably, our study found that many proteins with known antifibrotic functions were downregulated in DD. For example, the cartilage oligomeric matrix protein (COMP) has previously been shown to assist collagen secretion by skin fibroblasts, and it is important for the correct organisation of ECM fibres [48]. Also, proteoglycan 4 (PRG4) has been shown to function as an antifibrotic modulator in the joints, where it negatively regulates fibroblast differentiation into α-SMA^+^ myofibroblasts [49]. The results of the proteomic analysis presented in this report are consistent with previous studies [50,51,52,53] and provide notable evidence that DD shares common mechanisms with other fibrotic disorders.

Fibroblasts make up the main cell population in all stages of DD (both during nodule and cord formation), producing excess connective tissue components. The differentiation of fibroblasts into α-SMA-expressing myofibroblasts is considered to be responsible for the typical clinical symptoms [12]. At the same time, macrophages act as key regulators of tissue fibrosis by communicating with other immune and stromal cells through the production of various soluble factors [21,54]. We found that the ECM of DD tissue regulates the secretion of inflammatory cytokines IL-6 and TNF by macrophages. IL-6 and TNF, together with the fibrosis master regulator TGF-β, have been demonstrated to play crucial roles in inflammation, fibroblast activation, and collagen deposition—processes that contribute to wound healing and fibrosis development [12,22]. Interestingly, a similar study using a co-culture system of DD-derived myofibroblasts and THP-1 macrophages suggested that IL-6 and TNF secretion profiles are strongly influenced by paracrine macrophage–myofibroblast interactions [55]. This implies that the fibrotic ECM and DD-derived myofibroblasts can both play a crucial role in the immunoregulation of the pro-inflammatory milieu. It is currently unclear how the macrophage–myofibroblast physical interactions in the fibrotic tissue, shown by our study and more thoroughly investigated by Lodyga et al. [56], modify cytokine secretion in DD. Future work should include 3D co-culture systems to assess the paracrine signalling events between macrophages, myofibroblasts, and the ECM components to mimic the development of DD.

We could not detect a direct effect of the DD ECM on fibroblast proliferation or differentiation towards the α-SMA^+^ phenotype. However, our results showed that macrophages that had been cultured in DD ECM-containing medium increased the fibroblast proliferation rate as well as type I collagen production and α-SMA expression. Moreover, we found indications that DD ECM can directly support macrophage-to-myofibroblast transition, which could be another contributor to DD pathogenesis. In addition, we further showed that the DD ECM-stimulated macrophages were able to increase the fibroblast migration rate. These results indicate that ECM–macrophage interactions could be key steps in DD pathogenesis, supporting fibroblast migration into disease-affected tissue and, therefore, contributing to the expansion of fibrotic lesions, as well as cell differentiation into a profibrotic phenotype.

Current treatment strategies for DD include the surgical resection of the fibrotic palmar fascia [57], the collagenase-mediated dissolution of fibrotic contractures [58], the transdermal disruption of contracture cords (aponeurotomy) [59], and radiation therapy [60]. Fasciectomy techniques remain the gold standard in DD treatment, but their disadvantages include pain, long-term postoperative rehabilitation, and a high recurrence rate [61]. In addition to the methods listed, various non-surgical treatments, such as oral, intramuscular, topical, or intra-lesion steroids, have been proposed for the treatment of DD in the early stages [62]. However, none of them have shown the permanent prevention of disease recurrence, and some potentially effective treatments have been found to have a higher incidence of side effects. Therefore, DD is at present considered an incurable disease. As fibroproliferation is central to the pathogenesis of DD, growing efforts are being made to develop antifibrogenic pharmacotherapies [63]. Several medications already approved for antifibrotic therapy to treat other fibrotic diseases, such as TNF inhibitors and pirfenidone, have been investigated for use in DD [63]. Studies using mouse models of fibrosis have demonstrated that the inhibition or depletion of macrophages during the pathogenesis of lung, liver, and skin fibrosis [64,65,66,67] leads to antifibrotic effects and reduces scarring. Therefore, targeting dysregulated macrophage activation in fibrotic tissue holds promise for the development of novel antifibrotic therapies. Future studies could aim to elucidate the precise molecular mechanisms by which the specific dysregulated components of the fibrotic ECM contribute to the chronic coexistence and continued activation of macrophages and myofibroblasts in DD tissue, leading to fibrosis.

The main limitations of our study include the high variability in the degree of fibrosis in the collected DD tissue samples, and the biological variability between patients. Additionally, the sample size for the proteomic analysis was relatively small (*n* = 20). Another limitation is the absence of proper healthy control tissue, due to ethical challenges in acquiring palmar fascia from healthy individuals. Consequently, non-fibrotic palmar fascia (carpal tunnel surgery-derived) tissues were used. Furthermore, the in vitro experimental conditions, which were limited to two-dimensional monocultures, may impact the generalizability of the results to the complex pathophysiological processes occurring in diseased tissue.

## 4. Materials and Methods

### 4.1. Collection of Tissue Samples

DD tissue samples used in the experiments were obtained from surgically excised Dupuytren’s contracture nodular tissue from patients with extensive, stage 2 or 4 fibrosis of the palmar fascia. Control palmar fascia samples were obtained from patients unaffected by DD who underwent open carpal tunnel release surgery. The collection of tissue samples was approved by the Committee for Human Studies, University of Tartu (permit 335/T-1), and the procedures were performed in accordance with the Declaration of Helsinki. Written informed consent was obtained from all patients recruited to this study. A total of 20 DD and 20 control palmar fascia samples used for proteomic analysis was stored at −80 °C after collection until further processing. Appendix A shows data on the donor age and sex of the tissue samples used for proteomic analysis.

For immunofluorescence analysis, the nodular tissue was separated from the DD chords, embedded in O.C.T compound (Sakura Finetek Europe B.V., Alphen aan den Rijn, the Netherlands), and stored at −80 °C until further processing. Seven DD samples and seven control samples were used for immunofluorescence analysis (Appendix A).

### 4.2. Decellularization of Tissue Samples and Proteomic Analysis

To achieve decellularization, tissue samples were cut into pieces measuring 3–5 mm in size and incubated for 72 h in 0.5% sodium dodecyl sulphate and 1% Triton X-100 solution [68]. The solution was changed at 12 h intervals. Subsequently, the tissue pieces were washed in distilled water, initially for 2 × 30 min, followed by 2 × 2 h, and finally, for 4 h. Decellularized tissue samples were homogenized in lysis buffer consisting of 6 M guanidine hydrochloride, 100 mM Tris-HCl (pH 8.5), and 50 mM dithiothreitol for protein denaturation, alkalization, and reduction. The samples were subsequently heated at 95 °C for 10 min to deactivate proteases. Mass spectrometry-based proteomic analysis was conducted at the proteomics core facility at the University of Tartu, as previously described [69] (for more details, see the Appendix A). The mass spectrometry proteomic data have been deposited to the ProteomeXchange Consortium via the PRIDE [70] partner repository with the dataset identifier PXD059943.

The data analysis was performed using Perseus software version 1.6.15.0 [71]. For the quantitative analysis of the decellularized tissue proteome, the normalised spectral abundance factor (NSAF) was used [72]. The data were filtered to exclude proteins identified only through modified amino acids, and potential contaminants were removed from analysis. Proteins detected in at least 70% of the samples were retained in the data table. Missing values were imputed for each sample separately on the basis of a normal distribution with a width of 0.3 and a downshift of 1.8. To analyse differences in protein expression, the Student’s *t*-test was used, with a permutation-based false discovery rate (FDR) correction (FDR = 0.05) and 10 000 randomizations. Enrichment analysis was conducted using the ShinyGO v0.741 database (http://bioinformatics.sdstate.edu/go74/ (accessed on 10 March 2023)). For protein–protein interaction analysis, STRING version 11.5 (https://string-db.org (accessed on 10 March 2023)) with a high confidence threshold of 0.7 was used. To cluster upregulated and downregulated proteins, *k*-means clustering was performed, with the number of clusters set to 4 for upregulated proteins and 3 for downregulated proteins. Clusters were annotated using the ShinyGO v0.741 database.

### 4.3. Fibroblast Separation and Culture

The nodular tissue pieces from the DD cords (2–3 mm in size) were placed in tissue culture dishes and allowed to adhere for 5–7 min. Subsequently, 10 mL of fibroblast growth medium was added, and the tissue pieces were incubated at 37 °C and 5% CO_2_. The growth medium consisted of DMEM, 10% foetal bovine serum and 1% penicillin-streptomycin (100 U/mL and 0.1 mg/mL, respectively) (all from Life Technologies Corporation, Grand Island, NY, USA). After 7 days, the medium was replaced, and cell migration from the tissue fragments was assessed under light microscope. Once migration had reached a sufficient level, the tissue fragments were removed from the dish, and the cells were subcultured onto 60 mm dishes. The tissue fragments were stored on the dishes for a maximum of 2 weeks before subculturing. Fibroblasts from 4 patients with Dupuytren’s contracture were pooled together for experiments to minimise biological variation.

### 4.4. Monocyte Separation and Macrophage Culture

Buffy coat (obtained from the Tartu University Hospital Blood Center) was diluted 1:1 with DPBS containing 1 mM EDTA. Diluted buffy coat was layered on top of the Ficoll-Paque PLUS (Cytiva Sweden AB, Uppsala, Sweden) at a 1:3 ratio. The tubes were centrifuged at 400 rcf for 35 min at 23 °C, without the use of a brake. After centrifugation, the uppermost plasma layer was collected and inactivated in a water bath at 56 °C for 30 min. The plasma was then cooled on ice, centrifuged at 4000 rcf for 20 min, and filtered through a 0.22 μm filter.

Mononuclear cells were collected after removing the plasma layer. Cells were washed 3 times with DPBS containing 1 mM EDTA. When needed, erythrocyte lysis was performed after the first wash as an intermediate step, by incubating the cells in erythrocyte lysis buffer (BD Biosciences, Franklin Lakes, NJ, USA). Subsequently, the mononuclear cells were incubated in a 10 cm tissue culture dish at a density of 5 × 10^6^ cells/mL for 45 min. After incubation, the monocytes were allowed to adhere to the plastic, and all remaining cells were washed away with DPBS. A total of 2 × 10^6^ cells were reseeded on 35 mm tissue culture plates or 0.5 × 10^6^ cells per 24-well plate. Monocytes were cultured in RPMI 1640 (Biowest, Nuaille, France), supplemented with 10% heat-inactivated autologous plasma, 1% penicillin–streptomycin (100 U/mL and 0.1 mg/mL, respectively) (Life Technologies Corporation, Grand Island, NY, USA), and 50 ng/mL GM-CSF (PeproTech, Thermo Fisher Scientific, Cranbury, NJ, USA). After 6–7 days, monocytes differentiated into macrophages.

To investigate the effects of decellularized DD or control tissue ECM on macrophages or fibroblasts, the cells were stimulated with decellularized and homogenized patient tissue samples (7.5 mg tissue/mL) in growth medium for 48 h. As a control for macrophage differentiation, we used 20 ng/mL IFN-γ plus 10 ng/mL LPS toward M1-type macrophages and 20 ng/mL IL-4 plus 20 ng/mL IL-13 (all PeproTech, Thermo Fisher Scientific, Cranbury, NJ, USA) toward M2-type macrophages. For the negative control, M0 type macrophages, fresh growth media lacking any stimuli were added to the cells. After the macrophages were cultured in DD or control ECM-containing medium, we used this macrophage culture medium to stimulate the fibroblasts for 48 h before conducting a transwell migration assay or immunofluorescence analysis.

### 4.5. Immunofluorescence Analysis

Tissue samples were embedded in O.C.T. medium (Sakura Finetek Europe B.V., Alphen aan den Rijn, the Netherlands) and frozen in liquid nitrogen. Subsequently, 10 μm sections were cut using a cryostat microtome and mounted on glass slides. Tissue sections and cells grown on the coverslips were fixed with 4% paraformaldehyde solution and permeabilized in 0.2% Triton X-100 solution. Samples were blocked with a blocking solution consisting of 5% normal donkey serum (Sigma-Aldrich, Merck Group, Darmstadt, Germany) in PBS. Samples were then incubated with primary antibodies overnight at 4 °C and, subsequently, with fluorochrome-conjugated secondary antibodies. All the antibodies used in the experiments can be found in Appendix A. To visualise cell nuclei, samples were stained with DAPI (Thermo Fisher Scientific, Eugene, OR, USA) at a final concentration of 0.1 μm/mL. Images were captured with an Olympus IX81 CellR microscope (Olympus Corporation, Hamburg, Germany) using 10× objective and Hamamatsu Orca ER (Hamamatsu Photonics, Herrsching am Ammersee, Germany) camera. Images were processed using Hokawo 2.1 software (Hamamatsu Photonics).

The fluorescence quantification of the tissue samples was performed using ImageJ (version 1.52) [73], where the integrated density of the fluorescent signal was determined for at least five randomly selected fields of view for each biological sample. For cell culture experiment analysis, the integrated density of the fluorescent signal was determined, and DAPI signal intensity was further used to normalise the intensity of the positive signal. Subsequently, the obtained fluorescence intensity values were divided by the average fluorescence intensity of cells cultured under control stimulation conditions, to make the experimental results comparable. The percentage of proliferating cells was determined using antibody to the Ki-67 antigen; the integrated density of Ki-67 signal was divided by the DAPI signal.

### 4.6. RNA Separation and RT-qPCR Analysis

Total RNA was extracted from stimulated macrophages using NucleoSpin RNA mini kit (Macherey-Nagel, Düren, Germany), according to the manufacturer’s instructions. Reverse-transcription was conducted with a RevertAid First Strand cDNA Synthesis kit (Thermo Fisher Scientific Baltics UAB, Vilnius, Lithuania), according to the manufacturer’s instructions. qPCR analysis was performed using LightCycler^®^ 480 II (Roche Diagnostics, Basel, Switzerland). See Appendix A for a more detailed description.

### 4.7. ELISA

Macrophage culture medium was collected after 72 h of stimulation with decellularized and homogenized patient tissue samples. TNF, IL-1β, IL-6, and IL-10 concentrations were determined using ABTS ELISA development kits (all PeproTech, Thermo Fisher Scientific, Cranbury, NJ, USA), according to the manufacturer’s instructions.

### 4.8. Transwell Migration Assay

First, the macrophages were stimulated with decellularized DD or control tissue ECM suspensions in growth medium for 48 h. Next, the collected macrophage culture medium was added to the fibroblast culture for 48 h before starting the migration assay was started. A total of 1.5 × 10^4^ fibroblasts were seeded in serum-free medium on the upper membrane of the transwell chamber (6.5 mm transwell with 8.0 mm pore polycarbonate membrane insert, Corning Incorporated, Kennebunk, ME, USA). A migration assay was conducted for 24 h. Cells were then fixed and stained with 0.5% Coomassie Brilliant Blue G-250 (Sigma-Aldrich, Steinheim, Germany). The cells that did not migrate through the membrane were removed with a moist cotton swab. At least 3 images of each well were captured using a 10× objective lens of a Nikon Eclipse TS100 microscope (Nikon Instruments, Melville, NY, USA) equipped with a digital camera head (DS-Vi1, Nikon) and a stand-alone controller and display unit (DS-L3, Nikon). The number of migrating cells per each field of view was quantified using ImageJ software (version 1.54g) and compared to control.

### 4.9. Statistics

Unless otherwise specified, statistical significance was determined by one-way ANOVA followed by Dunnett’s post-test (multiple comparisons) or Student’s *t*-test (two groups). *p*-values < 0.05 were considered significant.

## 5. Conclusions

Our human tissue sample-based study has identified novel components of Dupuytren’s contracture ECM that help to explain how the persistent localised microinflammatory environment can support and reinforce the fibrosis progression. We have shown that macrophages directly orchestrate the fibrotic process by transmitting signals from the fibrotic ECM toward myofibroblast differentiation. Our findings suggest that the interactions between macrophages and ECM should be considered as targets for therapeutic strategies in the future, to treat and prevent fibrotic diseases.

## Figures and Tables

**Figure 1 ijms-26-03146-f001:**
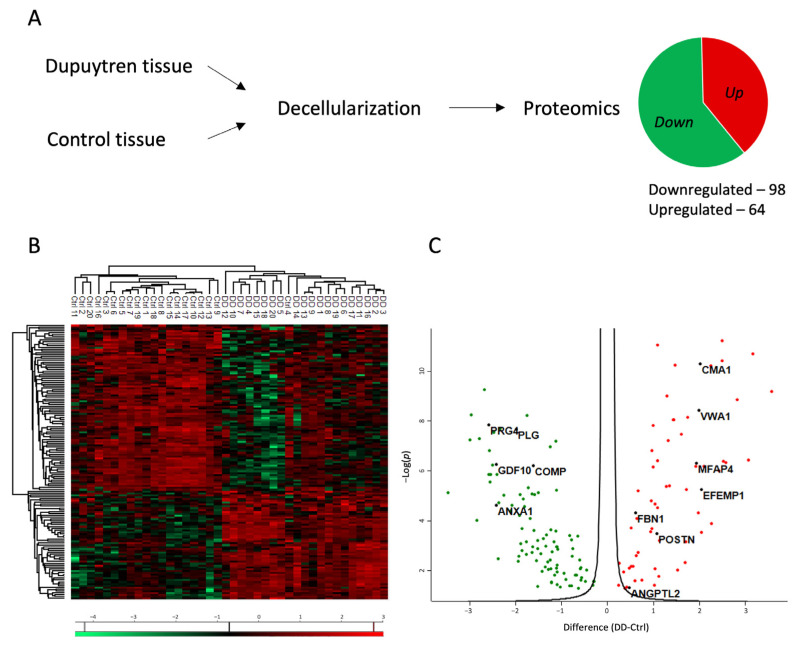
Proteomic profile of the changes in the ECM of Dupuytren’s contracture. (**A**) Experimental layout of the proteomic analysis. For more details, see the Section 4. (**B**) Heatmap of differentially enriched proteins in Dupuytren’s contracture and control palmar fascia tissue samples. (**C**) Volcano plot showing significantly up- and downregulated proteins in Dupuytren’s contracture ECM.

**Figure 2 ijms-26-03146-f002:**
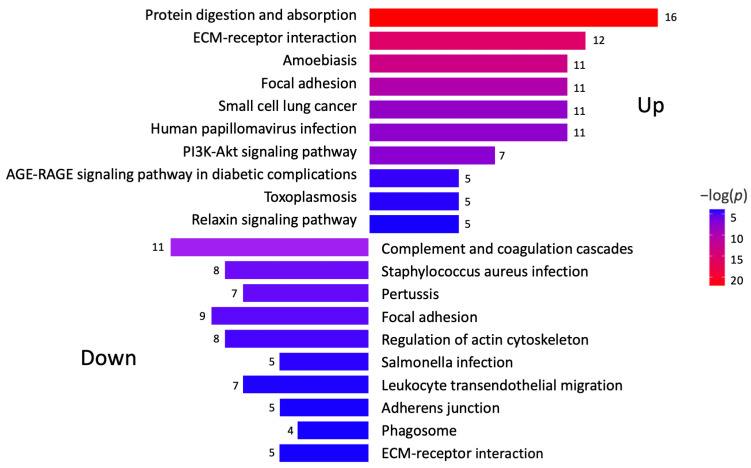
Analysis of signalling pathways potentially involved in DD. KEGG pathway analysis was performed to identify differentially expressed genes in Dupuytren’s contracture. Colour gradient shows statistical significance for each analysed pathway, and the numbers refer to genes involved in each signalling pathway.

**Figure 3 ijms-26-03146-f003:**
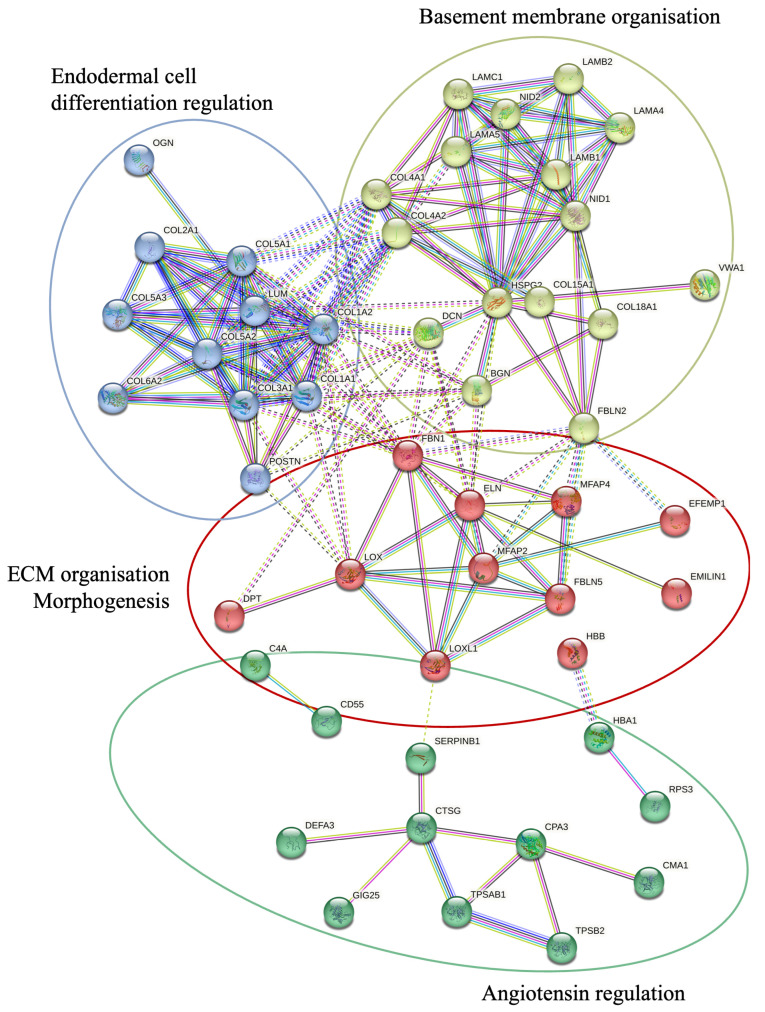
STRING protein–protein interactions between the proteins upregulated in DD. STRING network analysis of proteins with FC > 1.2 and *p*-value < 0.05 is shown. Four clusters were identified using a *k*-means approach, visualised in four different colours. The interactions between different clusters are illustrated with dotted lines, and intracluster interactions are shown with solid lines. The interactions shown are sourced from databases (light blue), experimental data (pink), textmining (green), co-expression (black), gene co-occurrence (dark blue), and protein homology (purple).

**Figure 4 ijms-26-03146-f004:**
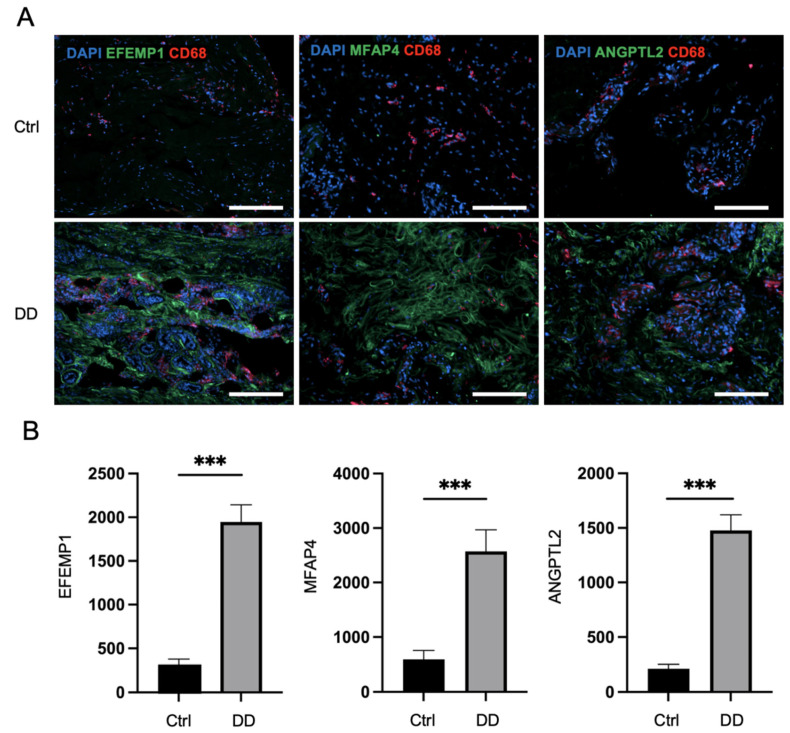
Immunofluorescence analysis of DD-associated markers in patient tissue sections. The representative samples (**A**) and relative quantification of the fluorescence signals of ANGPTL2, MFAP4, and EFEMP1 expression by mean integrated density (**B**) are shown. The scale bar is 200 μm. Results are presented as mean + SD; *n* = 7; *** *p* < 0.001.

**Figure 5 ijms-26-03146-f005:**
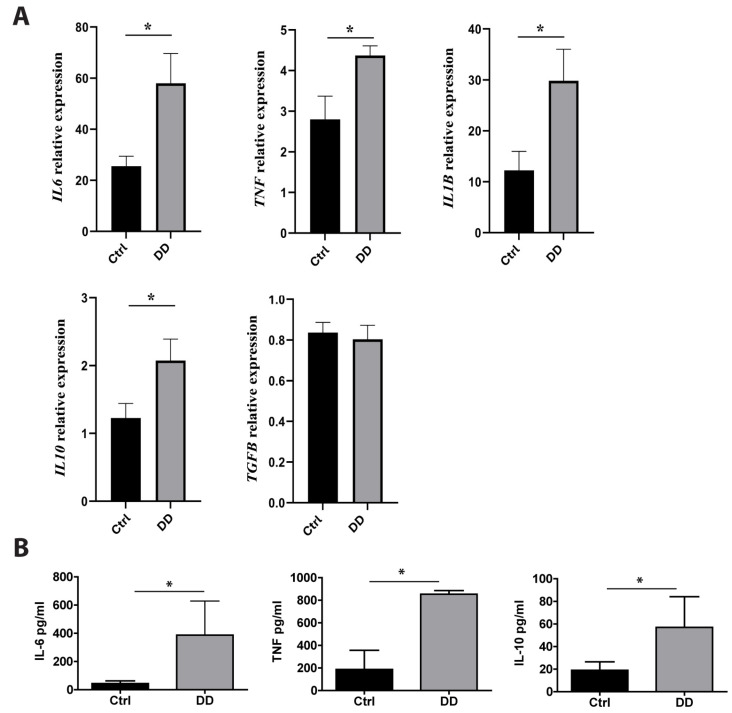
Effects of DD-tissue-derived ECM on macrophage cytokine production. Primary human macrophages were stimulated with decellularized homogenized DD or control palmar fascia ECM, and cytokine production was analysed. Relative RNA induction of *IL6*, *TNF*, *IL1B*, *IL10*, and *TGFB* was quantified using RT-qPCR (**A**), the graphs depict fold changes compared to unstimulated macrophages, *n* = 3–5. IL-6, TNF, and IL-10 protein concentrations in the cell culture medium were quantified using ELISA (**B**). *n* = 7; * *p* < 0.05. Results are presented as mean + SD.

**Figure 6 ijms-26-03146-f006:**
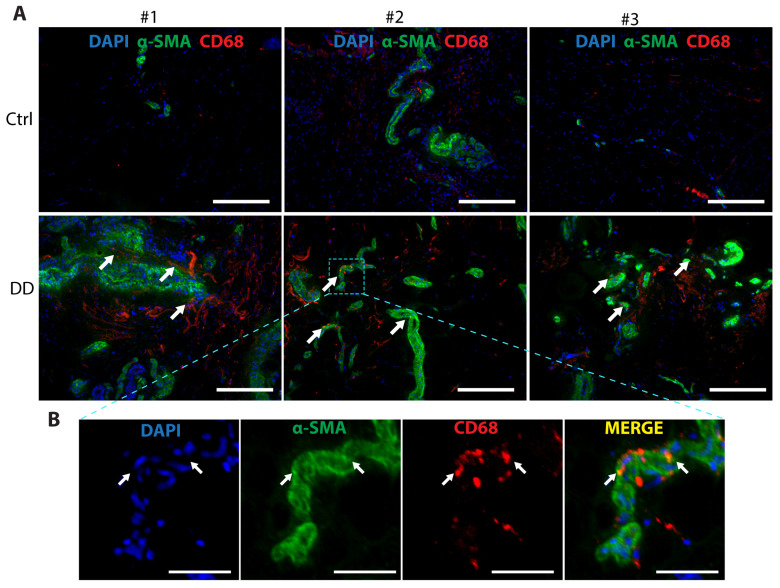
Co-expression of the myofibroblast marker α-SMA and the macrophage marker CD68 in DD palmar tissue sections. (**A**) Representative images of 3 DD and control patient tissue sections are shown. White arrows indicate areas of α-SMA and CD68 proximity in DD tissue samples. The scale bar is 200 μm. (**B**) The boxed region of the fibrotic sample from DD patient no. 2 is shown at higher magnification with channels separated and merged. Scale bar is 50 μm.

**Figure 7 ijms-26-03146-f007:**
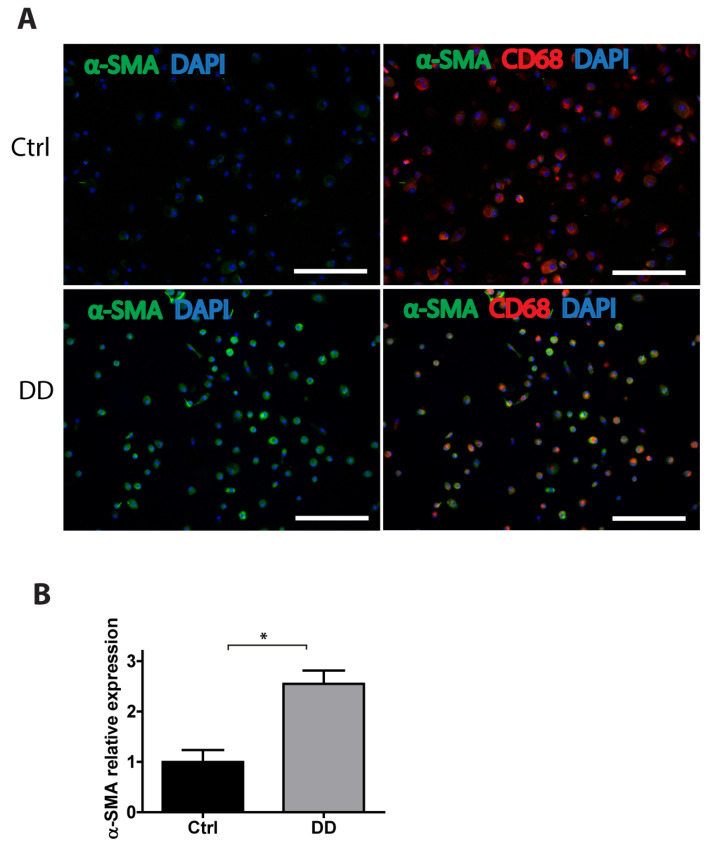
Exposure to DD tissue-derived ECM promotes MMT. Human monocyte-derived macrophages were stimulated for 72 h with decellularized homogenized DD or control palmar fascia ECM and stained for the myofibroblast marker α-SMA and the macrophage marker CD68. Representative images (**A**) and the quantification of 3 biological replicates (**B**) are shown. Results are presented as mean + SD and compared to control; * *p* < 0.05; the scale bar is 200 μm.

**Figure 8 ijms-26-03146-f008:**
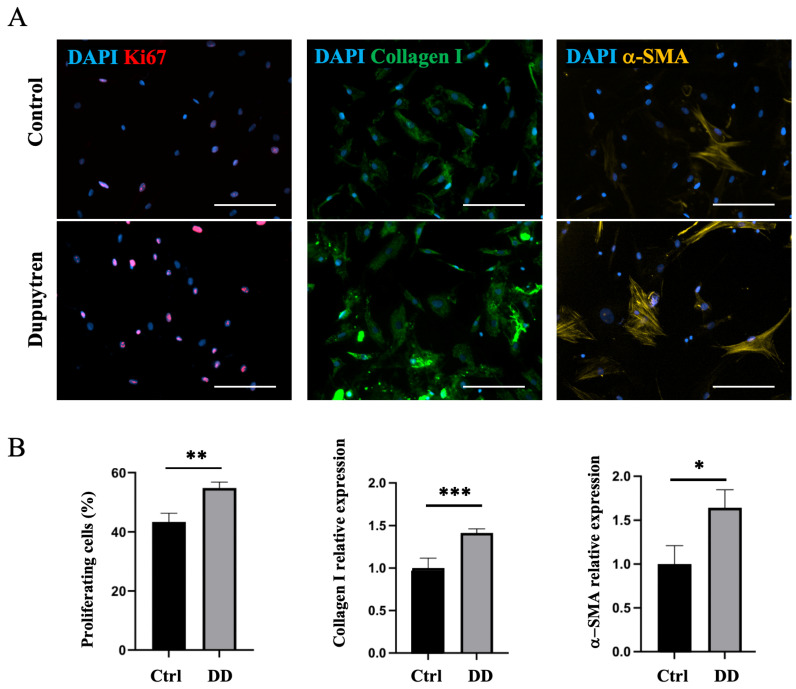
DD ECM-exposed macrophages support fibroblast proliferation and promote fibroblast differentiation into myofibroblasts. Human monocyte-derived macrophages were stimulated for 72 h with decellularized homogenized DD or control palmar fascia ECM, and subsequently, the cell culture medium was used to stimulate fibroblasts for 48 h (**A**). Cell proliferation, type I collagen, and α-SMA relative expression were quantified (**B**), *n* = 3. The scale bar is 200 μm. Results are presented as mean + SD and compared to control; * *p* < 0.05; ** *p* < 0.01; *** *p* < 0.001.

**Figure 9 ijms-26-03146-f009:**
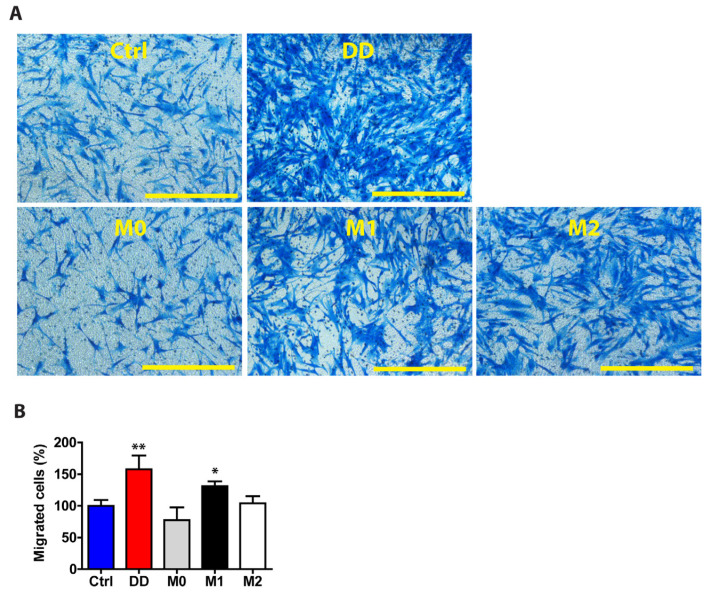
DD ECM-exposed macrophages signal to fibroblasts to promote their migration. Human monocyte-derived macrophages were stimulated for 72 h with control palmar fascia ECM (Ctrl), decellularized homogenized DD ECM, or M1/M2 cytokines as a control. Subsequently, the cell culture media were used to stimulate tissue-cultured fibroblasts for 48 h. Fibroblasts were allowed to migrate through a transwell chamber for 24 h. Representative images (**A**) and the quantitated results of the transwell assay (**B**). Results are presented as mean + SD and compared to Ctrl. Scale bar is 200 µm. *n* = 3; * *p* < 0.05; ** *p* < 0.01 compared to control palmar fascia ECM.

## Data Availability

Proteomic data are freely available at the ProteomeXchange Consortium via the PRIDE [70] partner repository with the dataset identifier PXD059943.

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
