# Peer review of "Pathological Changes in Extracellular Matrix Composition Orchestrate the Fibrotic Feedback Loop Through Macrophage Activation in Dupuytren’s Contracture"

_ijms, 2025, doi:10.3390/ijms26073146_

Round 1
Reviewer 1 Report
Comments and Suggestions for Authors
This is an interesting report describing proteomic, predicted signaling pathway, immunofluorescence and cell migration analyses of Dupuytren’s Disease (DD) cells and tissues compared to those from non-fibrotic palmar fascia (carpal tunnel surgery-derived). The methodologies employed are appropriate and the authors are to be commended for using non-fibrotic palmar fascia as their controls, rather than skin or other less physiologically appropriate tissues too often utilized by others conducting research in this area.
While commendable in many respects, the study does have some flaws that should be addressed before publication. The areas of concern are detailed below.
One is a failure to acknowledge relevant previous studies in this research area. For example, the authors claim to have “identified several novel profibrotic components upregulated in DD ECM and analysed the signalling pathways potentially involved in DD” (lines 95-97). These included CMA1, VWA1, MFAP4, EFEMP1, FBN1, POSTN, and ANGPTL2 (lines 118-121). While protein data for CMA, EFEMP1, FBN1 and ANGPTL2 are arguably novel, they are also consistent with several published reports indicating enhanced expression of the genes encoding these proteins, (see below), and such reports should be referenced. At a protein level, VWA1 has been previously identified in DD by Fede C et al, Int J Mol Sci. 2024 Jun 22;25(13):6865. doi: 10.3390/ijms25136865. PMID: 38999972; PMCID: PMC11241458; and POSTN has been very extensively studied at a protein level in DD (6 previous Pubmed reports in DD, including focused analyses by Vi L et al, Exp Cell Res. 2009 Dec 10;315(20):3574-86. doi: 10.1016/j.yexcr.2009.07.015. Epub 2009 Jul 18. PMID: 19619531; PMCID: PMC5017872. Similarly, several of the downregulated proteins identified by the authors have been previously identified at the gene expression level (see Shih B et al, J Hand Surg Am. 2009 Jan;34(1):124-36. doi: 10.1016/j.jhsa.2008.09.017. PMID: 19121738; and Satish L, et al, BMC Med Genomics. 2008 Apr 23;1:10. doi: 10.1186/1755-8794-1-10. PMID: 18433489; PMCID: PMC2377253). While I appreciate that the authors have identified these proteins using alternative methodologies, it is inappropriate for the authors to simply ignore previous gene expression reports that provide independent verification. All that is required is the addition of the words “consistent with previous studies” and the corresponding references.
I have several concerns regarding the interpretations of the immunofluorescence analyses.
Fig.4: As correctly stated by the authors, DD is a fibroproliferative disorder, and therefore nodular DD tissue sections contain substantially more fibroblasts and macrophages than non-fibrotic fascia. For Fig. 4, how were the immunofluorescence analyses corrected for differences in cell number between sections? The methods indicate that “the DAPI signal intensity was used to normalise the intensity of the positive signal for quantification of the cell culture samples. Subsequently, the obtained fluorescence intensity values were divided by the average fluorescence intensity of cells cultured under control stimulation conditions, to make the experimental results comparable.” (lines 447-450). Depending on the method, “DAPI signal intensity” corrects for variation in staining between experiments, but not necessarily for differences in cell number. I don’t doubt that EFEMP1, MFAP4 and ANGPTL2 immunoreactivity is greater in the DD samples, but how “mean integrated density” values (Fig. 4B) were achieved is quite unclear.
Fig. 5: These are interesting data. They provide an interesting and relevant comparison to similar recent studies, albeit by a different method, by Gonga-Cavé et al, Wound Repair Regen. 2021 Jul;29(4):627-636. doi: 10.1111/wrr.12928. Epub 2021 Jun 11. PMID: 34212454. Have the authors considered comparing (and referencing) the findings in this previous report? It would allow the authors speculate on ECM (this study) versus direct cellular effects (Gonga-Cavé et al) between cell types.
A minor nomenclature issue, but IL-6, TNF-a, IL-1b, IL-10 and TGF-b are protein names, not gene names. The figures in 5A need to be renamed IL6, TNFA, ILIB, IL10 and TGFB in accordance with HUGO nomenclature rules.
Fig. 6: Appreciating that MMT remains a controversial concept in fibrosis research, and that clearly demonstrating MMT is technically quite difficult, the data in Fig. 6 are nonetheless unconvincing as provided. Previous studies (notably Lodyga M et al, Sci Signal. 2019 Jan 15;12(564):eaao3469. doi: 10.1126/scisignal.aao3469. PMID: 30647145) have demonstrated that macrophages and myofibroblasts co-localize in fibrosis, and DD is unlikely to be an exception. The data provided here can just as easily be interpreted as cellular co-localization.
Fig.7: Why are the macrophages in Fig.7 not DAPI stained across all panels, as in Fig.8? That would allow a realistic appreciation of the cell numbers in the controls and DD samples. Providing a-SMA immunoreactivity alone in untreated control macrophages is expected to yield little or no signal. The reader cannot confirm that the same cell numbers were used across all experiments.
Fig.8: These images are difficult to interpret. The type-1 collagen immunoreactivity appears to be confined to the cell cytoplasm, despite type-1 collagen being a rapidly secreted ECM protein. No ECM collagen is evident, the cells are compact and do not look like myofibroblasts. When compared to the a-SMA immunoreactivity, the cell morphology is very different, resembling a more standard myofibroblast appearance, with clear evidence of actin stress fibers. If the cells in each panel underwent the same treatments (cell culture medium), why are the cell morphologies so different across panels?
Minor grammatical errors: line 21 of the abstract should read “were” (plural), not “was” (singular), some minor spelling errors (chords vs cords, line 391), etc.
Author Response
Comment 1: One is a failure to acknowledge relevant previous studies in this research area. For example, the authors claim to have “identified several novel profibrotic components upregulated in DD ECM and analysed the signalling pathways potentially involved in DD” (lines 95-97). These included CMA1, VWA1, MFAP4, EFEMP1, FBN1, POSTN, and ANGPTL2 (lines 118-121). While protein data for CMA, EFEMP1, FBN1 and ANGPTL2 are arguably novel, they are also consistent with several published reports indicating enhanced expression of the genes encoding these proteins, (see below), and such reports should be referenced. At a protein level, VWA1 has been previously identified in DD by Fede C et al, Int J Mol Sci. 2024 Jun 22;25(13):6865. doi: 10.3390/ijms25136865. PMID: 38999972; PMCID: PMC11241458; and POSTN has been very extensively studied at a protein level in DD (6 previous Pubmed reports in DD, including focused analyses by Vi L et al, Exp Cell Res. 2009 Dec 10;315(20):3574-86. doi: 10.1016/j.yexcr.2009.07.015. Epub 2009 Jul 18. PMID: 19619531; PMCID: PMC5017872. Similarly, several of the downregulated proteins identified by the authors have been previously identified at the gene expression level (see Shih B et al, J Hand Surg Am. 2009 Jan;34(1):124-36. doi: 10.1016/j.jhsa.2008.09.017. PMID: 19121738; and Satish L, et al, BMC Med Genomics. 2008 Apr 23;1:10. doi: 10.1186/1755-8794-1-10. PMID: 18433489; PMCID: PMC2377253). While I appreciate that the authors have identified these proteins using alternative methodologies, it is inappropriate for the authors to simply ignore previous gene expression reports that provide independent verification. All that is required is the addition of the words “consistent with previous studies” and the corresponding references.
Response 1: The necessary corrections have been made. We have acknowledged previous relevant studies and included the corresponding references in the Discussion.
I have several concerns regarding the interpretations of the immunofluorescence analyses.
Comment 2: Fig.4: As correctly stated by the authors, DD is a fibroproliferative disorder, and therefore nodular DD tissue sections contain substantially more fibroblasts and macrophages than non-fibrotic fascia. For Fig. 4, how were the immunofluorescence analyses corrected for differences in cell number between sections? The methods indicate that “the DAPI signal intensity was used to normalise the intensity of the positive signal for quantification of the cell culture samples. Subsequently, the obtained fluorescence intensity values were divided by the average fluorescence intensity of cells cultured under control stimulation conditions, to make the experimental results comparable.” (lines 447-450). Depending on the method, “DAPI signal intensity” corrects for variation in staining between experiments, but not necessarily for differences in cell number. I don’t doubt that EFEMP1, MFAP4 and ANGPTL2 immunoreactivity is greater in the DD samples, but how “mean integrated density” values (Fig. 4B) were achieved is quite unclear.
Response 2: We apologise for the misleading wording in the Methods section. The fluorescence quantification of the tissue samples was performed using ImageJ, where the integrated density of fluorescent signal was determined for at least five randomly selected fields of view for each biological sample. The quantification values presented in Fig. 4B are not corrected for DAPI signal intensity or cell number. We have now revised the wording in this part of the Methods section.
Comment 3: Fig. 5: These are interesting data. They provide an interesting and relevant comparison to similar recent studies, albeit by a different method, by Gonga-Cavé et al, Wound Repair Regen. 2021 Jul;29(4):627-636. doi: 10.1111/wrr.12928. Epub 2021 Jun 11. PMID: 34212454. Have the authors considered comparing (and referencing) the findings in this previous report? It would allow the authors speculate on ECM (this study) versus direct cellular effects (Gonga-Cavé et al) between cell types.
Response 3: We appreciate this very relevant suggestion. We have now extended the Discussion section to place our findings on macrophage secretion of inflammatory cytokines in the context of existing literature and included the suggested reference.
Comment 4: A minor nomenclature issue, but IL-6, TNF-a, IL-1b, IL-10 and TGF-b are protein names, not gene names. The figures in 5A need to be renamed IL6, TNFA, ILIB, IL10 and TGFB in accordance with HUGO nomenclature rules.
Response 4: The nomenclature issues have now been corrected.
Comment 5: Fig. 6: Appreciating that MMT remains a controversial concept in fibrosis research, and that clearly demonstrating MMT is technically quite difficult, the data in Fig. 6 are nonetheless unconvincing as provided. Previous studies (notably Lodyga M et al, Sci Signal. 2019 Jan 15;12(564):eaao3469. doi: 10.1126/scisignal.aao3469. PMID: 30647145) have demonstrated that macrophages and myofibroblasts co-localize in fibrosis, and DD is unlikely to be an exception. The data provided here can just as easily be interpreted as cellular co-localization.
Response 5: We have now improved Fig. 6 by adding a panel of images with higher magnification. This allows for a better evaluation of the cellular co-localization and overlap of the CD68 and α-SMA marker immunoreactivity within the same cell. The possibility of MMT occurring in DD is also supported by the results presented in Fig. 7. However, we agree that, in the future, the MMT concept should be investigated further using different approaches.
Comment 6: Fig.7: Why are the macrophages in Fig.7 not DAPI stained across all panels, as in Fig.8? That would allow a realistic appreciation of the cell numbers in the controls and DD samples. Providing a-SMA immunoreactivity alone in untreated control macrophages is expected to yield little or no signal. The reader cannot confirm that the same cell numbers were used across all experiments.
Response 6: We have now improved Fig. 7 by replacing the separated α-SMA channel images in the left panels with combined α-SMA + DAPI channel images. We hope that this allows easier interpretation of the data for the reader.
Comment 7: Fig.8: These images are difficult to interpret. The type-1 collagen immunoreactivity appears to be confined to the cell cytoplasm, despite type-1 collagen being a rapidly secreted ECM protein. No ECM collagen is evident, the cells are compact and do not look like myofibroblasts. When compared to the a-SMA immunoreactivity, the cell morphology is very different, resembling a more standard myofibroblast appearance, with clear evidence of actin stress fibers. If the cells in each panel underwent the same treatments (cell culture medium), why are the cell morphologies so different across panels?
Response 7: In our experiments, we found that the differentiation of fibroblasts to a myofibroblast-like phenotype was heterogeneous in the DD treated group. This is more evident in the α-SMA panel of Fig. 8, which more adequately reflects the changes in cell shape and size. Concordantly, type 1 collagen immunoreactivity was heterogeneous in the DD fibroblast population. Unfortunately, the presented type 1 collagen immunoreactivity image in the DD panel did not adequately reflect a typical field of view. To address this issue, we have now replaced the image in the type 1 collagen panel to better illustrate typical changes in cell morphology. Nevertheless, we didn’t detect substantial ECM collagen deposition in our experiments. This may be due to the experimental conditions, which involved culturing the fibroblasts on glass coverslips for relatively short periods of time (48 h), or the analysis methods, which didn’t enable visualization of the extracellular collagen.
Comment 8: Minor grammatical errors: line 21 of the abstract should read “were” (plural), not “was” (singular), some minor spelling errors (chords vs cords, line 391), etc.
Response 8: We have corrected the mentioned errors and thoroughly proofread the text. We hope that the manuscript has been improved in this regard.
Reviewer 2 Report
Comments and Suggestions for Authors
1.The article should further standardize the statistical analysis of data. Additionally, the visualization of the PPI network should be presented in a more standardized and rigorous manner.
2.The article lacks a discussion on the study's limitations. It is recommended to include the following aspects: (1) Whether the current study is limited by sample size or in vitro experimental conditions, which may affect the generalizability of the results; (2) Whether there are other unconsidered influencing factors, such as gene expression regulation or mechanical forces affecting the tissue microenvironment.
3.The study found that the extracellular matrix regulates the secretion of inflammatory cytokines by macrophages; a more detailed discussion of the relevant inflammatory pathways is recommended.
4.The study mentions that the extracellular matrix of Dupuytren’s contracture promotes fibroblast migration. However, it is necessary to specify the details of this effect and clarify whether experimental methods and quantitative metrics have been explicitly provided.
5.The study highlights macrophage-extracellular matrix interactions as potential therapeutic targets but lacks a discussion on existing treatment strategies.
6.The article could further discuss the similarities and differences between this study and existing literature. For example, have other studies reported the upregulation of MFAP4, EFEMP1, and ANGPTL2 in Dupuytren’s contracture? Additionally, do these proteins exhibit similar roles in other fibrotic diseases, such as pulmonary fibrosis or liver fibrosis?
7.The article should further discuss potential directions for future research to enhance its academic value and continuity.
Comments on the Quality of English LanguageThe article should provide a more detailed discussion on potential directions for future research to enhance its academic significance and ensure the continuity of the study.
Author Response
Comment 1: The article should further standardize the statistical analysis of data. Additionally, the visualization of the PPI network should be presented in a more standardized and rigorous manner.
Response 1: We have now improved the visualization of the PPI network (Fig. 3) and further standardized the statistical analysis of data. We have also made corrections to the Methods section and provided further clarifications in the figure legends where applicable. However, the modifications did not significantly alter the interpretation of the statistical analyses.
Comment 2: The article lacks a discussion on the study's limitations. It is recommended to include the following aspects: (1) Whether the current study is limited by sample size or in vitro experimental conditions, which may affect the generalizability of the results; (2) Whether there are other unconsidered influencing factors, such as gene expression regulation or mechanical forces affecting the tissue microenvironment.
Response 2: We have included a new paragraph in the Discussion that outlines the limitations of our study.
Comment 3: The study found that the extracellular matrix regulates the secretion of inflammatory cytokines by macrophages; a more detailed discussion of the relevant inflammatory pathways is recommended.
Response 3: We have now extended the Discussion section to place our findings on macrophage secretion of inflammatory cytokines in the context of existing literature.
Comment 4: The study mentions that the extracellular matrix of Dupuytren’s contracture promotes fibroblast migration. However, it is necessary to specify the details of this effect and clarify whether experimental methods and quantitative metrics have been explicitly provided.
Response 4: We have now improved Fig. 9 and further clarified the details of the fibroblast migration experiments and the statistical analysis. Most notably, we normalised the number of migrated cells to control to make the data presentation more comprehensible.
Comment 5: The study highlights macrophage-extracellular matrix interactions as potential therapeutic targets but lacks a discussion on existing treatment strategies.
Response 5: We have extended the Discussion to cover existing treatment strategies.
Comment 6: The article could further discuss the similarities and differences between this study and existing literature. For example, have other studies reported the upregulation of MFAP4, EFEMP1, and ANGPTL2 in Dupuytren’s contracture? Additionally, do these proteins exhibit similar roles in other fibrotic diseases, such as pulmonary fibrosis or liver fibrosis?
Response 6: We have extended the Discussion to further place our results in the context of existing literature. The Discussion now includes an overview of the roles of MFAP4, EFEMP1, and ANGPTL2 in fibrotic diseases. However, we found no previous reports describing MFAP4, ANGPTL2, or EFEMP1 expression in Dupuytren’s contracture.
Comment 7: The article should further discuss potential directions for future research to enhance its academic value and continuity.
Response 7: We have included a new paragraph in the Discussion that outlines the future directions.
Round 2
Reviewer 1 Report
Comments and Suggestions for Authors
Greatly improved, although I still find the MMT data less than convincing. Macrophages are typically ~ < 30 microns, whereas myofibroblasts are typically > 100 microns. Fig 6B still looks like macrophages associated with myofibroblasts to me, rather than the same cell. The alpha SMA positive cells presumed to be macrophages in Fig. 7 are interesting, but I would need more convincing proof that they are "pre-myofibroblasts" than just some alpha SMA immunoreactivity. However, I respect that the authors are entitled to their own conclusions based on these data.
Only minor edit noted was the use of TNF vs TNF-alpha, as both terminologies are used (lines 358-371). I suggest switching to TNF throughout, since TNF alpha is old terminology and TNF beta was renamed lymphotoxin alpha many years ago, thus making the addition of alpha to TNF redundant. TNF is the more modern and correct nomenclature IMO.
Author Response
Comment 1: Greatly improved, although I still find the MMT data less than convincing. Macrophages are typically ~ < 30 microns, whereas myofibroblasts are typically > 100 microns. Fig 6B still looks like macrophages associated with myofibroblasts to me, rather than the same cell. The alpha SMA positive cells presumed to be macrophages in Fig. 7 are interesting, but I would need more convincing proof that they are "pre-myofibroblasts" than just some alpha SMA immunoreactivity. However, I respect that the authors are entitled to their own conclusions based on these data.
Response 1: Thank you for the insightful feedback.
Comment 2: Only minor edit noted was the use of TNF vs TNF-alpha, as both terminologies are used (lines 358-371). I suggest switching to TNF throughout, since TNF alpha is old terminology and TNF beta was renamed lymphotoxin alpha many years ago, thus making the addition of alpha to TNF redundant. TNF is the more modern and correct nomenclature IMO.
Response 2: The nomenclature issues have now been corrected. We have switched to using TNF throughout the manuscript and updated the labelling in the graphs in Fig 5.
Reviewer 2 Report
Comments and Suggestions for Authors
Considering that much of the content has been revised and supplemented, the manuscript can be received in its current form
Author Response
Comment 1: Considering that much of the content has been revised and supplemented, the manuscript can be received in its current form
Response 1: Thank you for the insightful remarks and suggestions, which greatly contributed to improving the quality of the manuscript.